# Gut Microbiota Dysbiosis in Diabetic Retinopathy—Current Knowledge and Future Therapeutic Targets

**DOI:** 10.3390/life13040968

**Published:** 2023-04-07

**Authors:** Dragos Serban, Ana Maria Dascalu, Andreea Letitia Arsene, Laura Carina Tribus, Geta Vancea, Anca Pantea Stoian, Daniel Ovidiu Costea, Mihail Silviu Tudosie, Daniela Stana, Bogdan Mihai Cristea, Vanessa Andrada Nicolae, Corneliu Tudor, Andreea Cristina Costea, Meda Comandasu, Mihai Faur, Ciprian Tanasescu

**Affiliations:** 1Faculty of Medicine, “Carol Davila” University of Medicine and Pharmacy, 020021 Bucharest, Romania; dragos.serban@umfcd.ro (D.S.);; 2Fourth Surgery Department, Emergency University Hospital Bucharest, 050098 Bucharest, Romania; 3Ophthalmology Department, Emergency University Hospital Bucharest, 050098 Bucharest, Romania; 4Faculty of Pharmacy, “Carol Davila” University of Medicine and Pharmacy, 020021 Bucharest, Romania; 5Faculty of Dental Medicine, “Carol Davila” University of Medicine and Pharmacy, 020021 Bucharest, Romania; 6Department of Internal Medicine, Ilfov Emergency Clinic Hospital, 022113 Bucharest, Romania; 7“Victor Babes” Infectious and Tropical Disease Hospital, 030303 Bucharest, Romania; 8Faculty of Medicine, Ovidius University Constanta, 900470 Constanta, Romania; 9General Surgery Department, Emergency County Hospital Constanta, 900591 Constanta, Romania; 10Department of Nephrology, Diaverum Clinic Constanta, 900612 Constanta, Romania; 11Faculty of Medicine, University “Lucian Blaga”, 550169 Sibiu, Romania; 12Department of Surgery, Emergency County Hospital Sibiu, 550245 Sibiu, Romania

**Keywords:** diabetic retinopathy, gut microbiota, dysbiosis, LPS, SCFAs, blood retinal barrier, apoptosis

## Abstract

Diabetic retinopathy is one of the major causes of blindness today, despite important achievements in diagnosis and therapy. The involvement of a gut–retina axis is thought to be a possible risk factor for several chronic eye disease, such as glaucoma, age-related macular degeneration, uveitis, and, recently, diabetic retinopathy. Dysbiosis may cause endothelial disfunction and alter retinal metabolism. This review analyzes the evidence regarding changes in gut microbiota in patients with DR compared with diabetics and healthy controls (HCs). A systematic review was performed on PubMed, Web of Science, and Google Scholar for the following terms: “gut microbiota” OR “gut microbiome” AND “diabetic retinopathy”. Ultimately, 9 articles published between 2020 and 2022 presenting comparative data on a total of 228 T2DM patients with DR, 220 patients with T2DM, and 118 HCs were analyzed. All of the studies found a distinctive microbial beta diversity in DR vs. T2DM and HC, characterized by an altered Firmicutes/Bacteroidetes ratio, a decrease in butyrate producers, and an increase in LPS-expressing and pro-inflammatory species in the Bacteroidetes and Proteobacteria phyla. The probiotic species Bifidobacterium and Lactobacillus were decreased when compared with T2DM. Gut microbiota influence retinal health in multiple ways and may represent a future therapeutic target in DR.

## 1. Introduction

Diabetes mellitus is a major health problem globally, accounting for 537 million patients worldwide, a number that is expected to reach 700 million by 2045 [1,2]. Insulin resistance and low-grade systemic inflammations lead to multiple organ damage, which is brought about by microvascular and macrovascular complications. Despite significant achievements in early diagnosis and therapy, diabetic retinopathy remains the leading cause of blindness in the working-age population and has a severe impact on patients’ quality of life. Epidemiologic studies have found that within 20 years following the onset of DM, almost all patients with type 1 DM (T1DM) and over 60% of those with type 2 DM (T2DM) will present retinal damage [3,4,5].

The changes attributed to hyperglycemia in the retina are due both to the effects on the microcirculation and the retinal cells. The accumulation of subendothelial advanced glycation end products at the level of the extracellular matrix leads to the alteration of intercellular junctions, favoring the extravascular migration of inflammatory cells and the activation of monocytes and macrophages via nuclear factor (NF)-κB. Additionally, the accumulation of highly reactive oxygen species (ROS) secondary to mitochondrial and endoplasmic reticulum dysfunction induced by hyperglycemia leads to the accumulation of membrane peroxidation products and DNA damage, with a cytotoxic effect not only on the retinal Muller cells, astrocytes, and photoreceptors, but also at the capillary level. The caspase activation pathways are implicated in the induction of apoptosis [6], and inflammation is activated via NF-kB, with increased vascular permeability, leakage, ischemia, and the expression of VEGF [7,8]. Local and systemic inflammation plays an important role in the pathology of diabetic retinopathy, leading to endothelial dysfunction, vascular leakage, and damage of neuroretina cells [9].

Multiple studies have reported a gut–eye axis and concluded that slight changes in gut microbiota may significantly influence ophthalmological diseases, such as uveitis, age-related macular degeneration, and glaucoma [10,11,12]. The intestinal microbiota is essential in maintaining the integrity of the intestinal walls, preventing the development of pathogenic germs, and also maturing immunity. Along with facilitating the absorption of nutrients, the microbiota is responsible for the production of some metabolites and signaling molecules that modulate multiple metabolic reactions in the host organism [13,14,15].

The gastrointestinal ecosystem is extremely rich, diverse, and dynamic. In light of recent findings, it tends to be considered more like another organ in our body due to its complex interactions with our metabolism. More than 5000 different species live in the human gut, which includes up to 10^14^ cells, 10 times more than the total number of cells in our body [16,17]. However, most of them are non-cultivable by traditional laboratory techniques. The development of new methods of identification, based on 16S rRNA gene sequence analysis, allows a better approach to exploring and describing the diversity of gut microbiota in different healthy or pathologic conditions. A decrease in the taxonomic diversity of gut microbiota was reported in patients with cancer and other chronic inflammatory diseases [18,19].

Recently, gut microbiota were investigated as a potential risk factor for diabetic retinopathy, but also as a potential therapeutic tool to prevent microvascular complications in patients with T2DM. The present review aims to document the evidence regarding the specific changes in gut microbiota in DR patients.

## 2. Materials and Methods

We performed a systematic review by searching the international databases PubMed/Medline, Google Scholar, and Web of Science using the term “diabetic retinopathy” AND “gut microbiota” OR “gut microbiome”. All original articles in the English language, published before January 2023, that reported data regarding gut microbiota in patients with type 2 diabetes and diabetic retinopathy were screened. A manual search of the references of the relevant reviews on this topic was performed. For potentially relevant papers, full-text articles were obtained, and the inclusion and exclusion criteria were applied.

### 2.1. Inclusion and Exclusion Criteria

The inclusion criteria were as follows: (1) articles reporting comparative results in terms of alfa and beta diversities between diabetic retinopathy subjects and healthy controls or diabetic subjects without retinopathy; (2) articles in which the subjects included in the study did not take antibiotics, probiotics, or prebiotics in the month before fecal sampling.

Exclusion criteria were as follows: (1) insufficient documentation of the study group; (2) studies that described a certain intervention on gut microbiota; (3) conference abstracts, letters to editors, and editorials, due to limited data. For potentially relevant papers, full-text articles were obtained, and the inclusion and exclusion criteria were applied.

### 2.2. Data Extraction and Analysis

All articles were screened by title and abstract by two independent reviewers, in accordance with the Oxford Centre for Evidence-Based Medicine. A PRISMA flowchart was employed to screen the papers for eligibility. Furthermore, we used the population, intervention, comparison, outcome, and study design (PICOS) strategy to guide our study rationale and to make a clear systematic literature search:

P (patients): type 2 diabetic patients with diabetic retinopathy (DR), proven by ophthalmological exam in accordance with the International Clinical Diabetic Retinopathy Disease Severity Scale.

I (intervention): gut microbiota analysis from a fecal specimen.

C (comparison): healthy controls and/or diabetic subjects without retinopathy.

O (outcomes): measured in terms of alfa and beta diversity analysis. Changes at phylum, genera, and species levels in the gut microbiome were documented.

S: cohort prospective studies with comparative matched groups in terms of age, sex, BMI, and diabetes duration (for the diabetic group without retinopathy) were included in this review.

An assessment of bias was performed using the Newcastle–Ottawa Scale [20], and a score ≥ 5 indicated adequate quality for inclusion in the present review. Any disagreement between two reviewers regarding the data collection or quality assessment was solved by discussion or by an additional author referring to the full text of the study in question.

General data regarding the patients included in the comparative analysis were taken from the tables and texts of the articles as mean and standard deviation (SD). The statistical analysis was performed using the SciStat^®^ software (MedCalc Version 20.218 Software Ltd., Ostend, Belgium).

Due to the heterogeneity and incomparability of processing and sequencing methodology, e.g., 16S rRNA or metagenomic sequencing, a meta-analysis was not conducted.

## 3. Results

Ultimately, 8 articles published between 2020 and 2022 presenting comparative data on a total of 204 T2DM patients with diabetic retinopathy (DR), 199 patients with T2DM, and 88 healthy controls (HC) were retained for the systematic review. A flowchart of database screening and article selection is presented in Figure 1.

### 3.1. General Characteristics of the Included Studies

According to the Newcastle–Ottawa Scale, two studies were evaluated with a score of 6 (2/8), six studies were evaluated with a score of 7 (5/8), and one study was evaluated with a score of 8 (1/8), indicating the relatively high quality of the studies selected. 

All of the papers were prospective cohort studies which compared gut microbiota in a selected group of DR patients with a comparison group of T2DM patients and/or healthy controls (HC), matched in terms of age and sex distribution. The comparison group was matched for BMI in six (6/8) of the studies, thus excluding the possible impact of obesity on the gut microbiome, a possible source of bias. When a microbiome analysis was compared with a T2DM group, the duration of diabetes for the T2DM group was matched in six out of seven studies, and HbA1C values were matched in five out of seven studies (Table 1).

None of the participants took antibiotics, probiotics, or prebiotics for at least one month before fecal sampling. In all of the studies, patients aged below 18 years, patients with a previously known serious chronic disease or infection, and patients with unhealthy habits (increased alcohol consumption, smoking) were excluded. 

The proportion of patients that underwent metformin therapy at the moment of stool sample harvesting was reported in five out of eight studies, and it varied widely (between 10.81% [27] and 100% [21,24]). However, in each case the authors reported matched percentages to minimize the effects of metformin therapy on gut microbiota as a possible source of errors.

### 3.2. Richness and Diversity Changes in the Gut Microbiome in DR vs. T2DM and HC

In eight studies, gut microbiota were analyzed in terms of their richness and diversity of species for the DR group vs. the T2DM and HC groups, and this was followed by a comparative assessment of the relative abundance of gut bacteria and their composition via microbial taxon assignment at both phylum and family levels [18,19,21,22,23,24,25,26]. 

DNA Extraction and 16S rRNA gene sequencing was the preferred technique (used in seven (7/8) studies [18,19,21,22,23,24,26]), while shotgun metagenomic sequencing was used in only one [25].

Alpha diversity indices evaluate “within sample” diversity in terms of richness and evenness. Richness refers to the number of types of microorganisms in a sample, while evenness characterizes the uniformity of distribution of the population from each species in a sample. In the studies included in this review, richness was evaluated based on two or more of the following specific indices: OTUs (operational taxonomy units), ACE (abundance based on coverage estimates), the observed species index (Sobs index), and Chao 1, a non-parametric method for estimating the number of species in a community. The Shannon index is an estimator of both species richness and evenness, but it weights richness. The Simpson index and invSimpson index are indicators of diversity, reflecting the probability of two microorganisms taken from random samples being from different species [27,28].

When analyzing the alpha diversity of gut microbiota in the DR vs. HC groups, most studies found no significant changes in richness [19,22,23,24]. In the study by Zhou et al. [21], OTUs and the Chao 1 index were significantly decreased between both the DR and DM groups and between the DR and HC groups, suggesting differences in the number of microbial communities among the three compared groups. However, Huang [19] found a decreased invSimpson and Heip evenness index, while Das et al. [23] found a significant decrease in the Shannon index (Table 2).

When beta diversity was analyzed, all of the studies found significant differences in microbial composition between the communities, both when the DR group was compared with the healthy controls, and when compared with the T2DM group (Table 2 and Table 3). 

### 3.3. Changes at Phylum, Genera, and Species Levels 

In all of the groups, the most abundant phyla were Firmicutes, Bacteroidetes, Actinobacteria, Proteobacteria, and Verrucomicrobia. In each study, microbiome-specific changes were noticed in the DR group when compared with the T2DM and HC groups. 

When examining taxonomic distribution at the phylum level, three studies (3/8) found significantly decreased Firmicutes, increased Bacteroidetes, and a decreased F/B ratio in the DR group when compared with the HC group [19,22,24]. When compared with the T2DM group, the changes were less remarkable, except in the case of the study by Huang et al. [19]. Li et al. found only a decrease in Bacteroidetes, while Khan [26] observed a decrease in the F/B ratio. Li et al. [25] found that Actinobacteria were depleted in the DR group. Moreover, Verrucomicrobiota, Desulfobacterota, and Synergestota were increased in the DR group when compared with the HC group [22] (Table 4).

A comparative analysis at the genera and species level revealed multiple significant changes between the DR group and both the HC and T2DM groups. All of the studies concluded that diabetic retinopathy is associated with gut dysbiosis, which may impact various mechanisms of the retinal and endothelial cells. A quantitative analysis is difficult to perform due to the high diversity of species included in the reports, the multiple sources of bias related to the relatively limited number of cases analyzed, and the influence of diet and medication, such as metformin. However, in a qualitative analysis, all of the studies indicated a subtle disturbance in the gut equilibrium of pro- and anti-inflammatory bacteria, with multiple metabolic implications.

In DR patients, a decrease in SCFA-producing species with anti-inflammatory and metabolic modulating properties, such as *Roseburia* [18,19,21,24], *Faecalibacterium* spp. [18,19,21,23], Eubacterium_hallii_group [19,22], and *Blautia* [19,22,24], was reported in several studies (Table 4). However, other SCFA producers, such as *Romboutsia*, *Megamonas*, and *Parabacteroides* were found by Bai [22] and Das [23] to be increased in the DR group.

Moreover, mucin-degrading bacteria, which are associated with increased gut permeability and microbial translocation, were reported to be increased in the DR group. Bai et al. [22] found a higher level of Ruminococcus_torques_group in the DR group compared with that in the HC group, while *Akkermansia muciniphila* and Akkermansiaceae were reported to be increased in the DR group both when compared with the HC group [21,23] and the T2DM group [23].

Multiple species of Bacteroidetes and Proteobacteria phyla were found to be increased in the DR group, most of them—such as *Prevotella*, *Burkholderiacea*, *Muribaculacea*, *Alistipes*, *Bacteroides*, or pathogens such as *Escherichia* and *Enterobacter* [19,21,22,23,24,25]—being associated with systemic inflammation and LPS endotoxemia,.

Das [23] and Li [25] found a decreased *Bifidobacterium* population in the feces of patients with DR compared with that found in T2DM patients. When compared with HC patients, the results were conflicting. Das et al. [23] reported a decrease, while Huang et al. [19] reported an increased level. These findings could be partially related to metformin-induced dysbiosis in the diabetic patients.

## 4. Discussion

The reviewed studies showed an imbalance between the pro- and anti-inflammatory bacteria that compound the gut microbiome. The observed changes were subtle, and dysbiosis was evidenced more by a decrease in the anti-inflammatory species, such as a butyrate producer in Firmicutes phylum, Lactobacillus, or Bifidobacterium, with an altered F/B ratio. In a study by Prasad et al. [46], intermittent bacteriemia from the gut microbiome was evidenced, and microorganisms reached the plasma and retina, possibly due to an altered blood–retina barrier in the diabetic mice with DR. Thus, it would appear that the altered balance between the pro- and anti-inflammatory gut microbiome and the presence of pathogenic organisms could influence the status of DR. However, gut dysbiosis may be either a cause or a consequence of the underlying metabolic pathology.

### 4.1. SCFA Producers

SCFA production at the intestinal level occurs in a dynamic balance, being influenced by diet and by the relative proportion of the species involved, which belong to both Firmicutes and Bacteroidetes. The WHO recommends an optimal daily intake of 25 g of fiber [47]. Western diets are associated with lower levels of fiber, resulting in the absorption of most nutrients in the duodenum and proximal gut. Consequently, very few nutrients reach the proximal colon, and this affects the SCFA-producing population [48]. 

Besides the energy substrate they represent, SCFAs modulate the transmission of important biological signals.

Recent studies by Desjardin et al. [49] and Dewanjee et al. [50] have found that SCFAs mediate the inhibition of HDAC and restore retinal epithelial function in hyperglycemia by improving RPE fluid transport and blocking VEGF signaling. Chen et al. injected SCFAs intraperitoneally in an animal experimental model and proved that SCFAs may pass the blood–retinal barrier and reduce the production of inflammatory mediators via LPS-stimulated retinal astrocytes [51]. Their study found that SCFAs can inhibit IL-6, TNF-α, and the chemokines CXCL1 and CXCL12 in response to in vitro inflammatory stimuli such as the ligands of TLRs or IL-17, with butyrate being the most and acetate the least effective agent. Moreover, the migration of immune cells such as monocytes and macrophages induced by LPS decreased after SCFA administration [51]. However, different combinations of SCFAs may impact the retinal microenvironment differently, depending on health status or ongoing pathological processes. Chen S et al. [52] found that SCFA supplementation had little or no impact on healthy mice, but that it may aggravate retinal damage, and it resulted in inflammation in an experimental model of IOP-induced astrocyte activation. 

In the reviewed studies, SCFA producers, and especially butyrate producer species (*Roseburia*, *Faecalibacterium* spp., *Ruminococcaceae*, Eubacterium_hallii_group, *Streptococcus*, and Veillonaceae, Peptostreptococcaceae) were decreased in DR group when compared with the HC and T2DM groups. However, other acetate producers, such as *Romboutsia* [22], *Megamonas* [22], and Oscillospiraceae [19], were increased in DR patients and could be further tracked as early diagnosis biomarkers. Several studies found a positive correlation between *Megamonas* and blood glucose, serum fructosamine, duration of diabetes, glycated hemoglobin, and older age [37,53]. *Romboutsia* and Oscillospiraceae were correlated with an obesity-related genus, lipogenesis, and increased BMI [54,55,56]. Other neurodegenerative diseases, such as autism, depression, age-related macular degeneration, and Parkinson’s disease, were correlated with higher Oscillospiraceae [33,34,57,58]. 

### 4.2. Akkermansiaceae and A. muciniphila

In the current review, two studies [21,24] found higher levels of *A. muciniphila* in DR patients than in T2DM and HC patients, while five studies found no relevant differences. In animal studies, no positive correlation was found between *A. muciniphila* and the severity of retinal lesions [46].

The subject of more than 1300 papers since it was discovered in 2004, *A. muciniphila* belongs to *Verrucomicrobia phyla*, and it has been considered both beneficial and detrimental. Being a mucin-degrading bacteria, it was associated with disrupting the intestinal barrier, bacterial translocation, and inflammation in the context of “leaky gut” syndrome [21,26,59]. However, it seems also to have a role in mucin synthesis through an autocatalytic process [60]. Moreover, many studies have reported an inverse correlation between high levels of *A. muciniphila* and BMI, visceral adiposity, inflammation markers, and insulin resistance. Moreover, a prebiotic study by Everard et al. [61] found that an inulin-supplemented diet in obese and diabetic mice was associated with increased levels of Akkermansia and that it ameliorated some features of cardiometabolic syndrome. A further study by Depommier et al. [62] in volunteer humans found that both pasteurized and alive *A. muciniphila* were well tolerated and improved insulin sensitivity while reducing insulinemia and plasma total cholesterol [62,63].

### 4.3. Tauroursodeoxycholic Acid (TUDCA)-Producing Species 

Several species were found to have the ability to degrade primary biliary acids and to produce secondary biliary acids, such as Tauroursodeoxycholic acid (TUDCA) and Taurochenodeoxycholic Acid (TCDCA). Parabacteroides and species of the Firmicutes phylum, such as *Blautia*, were positively correlated with higher levels of TUDCA, a farnesoid X receptor (FXR) antagonist which regulates the glycolipid metabolism via its receptor for FXR and its bile acid G-protein-coupled membrane receptor (TGR5) [64]. Moreover, TGR5 was also evidenced in the retinal ganglion cell layer, and its activation has an anti-inflammatory and trophic effect. Beli et al. [65] found in an experimental animal model that higher TUDCA levels are associated with decreases in the number of acellular capillaries, the level of TNF alfa in the retina, and the abundance of inflammatory cells, such as CD11b+ macrophages, CD45+ leukocytes, and activated IBA-1+ microglia within the retina [65]. Several studies have found that TUDCA has multiple beneficial effects, e.g., the inhibition of photoreceptor apoptosis through a lessening of endoplasmic reticulum stress [59], the inhibition of the proliferation of the endothelial cells in the retina (protecting against neovascularization in DR) [59], and vascular repair via the recruitment of endothelial progenitor cells [66]. Murase et al. found that TUDCA activates MerTK, which is important for the phagocytosis of the photoreceptor outer segment in the elimination and renewal process, and they consider it a possible novel therapeutic agent in retinal neurodegenerative diseases.

Among the TUDCA-producing species, Parabacteroides were found to be increased in DR patients when compared with HC patients [23], while *Blautia* was found to be decreased in DR patients when compared with T2DM patients [19], but both increased [19] and decreased [22,23] when compared with HC patients. *Blautia* is also an acetate producer, providing the benefits of SCFA producers, such as inhibiting the effects of insulin at the level of the adipocyte via GPR41 and GPR43 [19]. Due to all of these potentially beneficial effects, higher levels observed in T2DM patients could prove to play a protective role against DR in hyperglycemic patients.

### 4.4. Probiotics and DR

Bifidobacterium and lactic acid bacteria are the strains most commonly used as probiotics due to their well-documented beneficial effects upon health. Both species are associated with immunomodulatory and antioxidant effects, decreased blood sugar, and improve insulin sensitivity. Lactic acid bacteria have been proven to promote a partial maturation of dendritic cells via the secretion of anti-inflammatory cytokines, and to promote a shift from M1 to M2 macrophages. In ophthalmic pathology, the potential benefits of *Lactobacillus* spp. have been investigated in experimental studies [30,63,67,68]. Iovieno et al. [67] found that administration of a topic *Lactobacillus acidophilus* diluted in saline solution (2 × 10 [8] CFU/mL) decreases inflammatory signs and symptoms in human patients with vernal keratoconjunctivitis [67]. In an experimental model, oral administration of a mixture of five probiotics consisting of *Lactobacillus casei*, *Lactobacillus acidophilus*, *Lactobacillus reuteri*, *Bifidobacterium bifidum*, and *Streptococcus thermophilus* showed promising results for treating autoimmunity in patients with uveitis and dry eye [63].

An experimental study on an age-related macular degeneration balb/c mice model found that a diet supplemented with *L. paracasei* KW3110 showed beneficial anti-inflammatory and neurotrophic effects. The *L. paracasei* KW3110 induced retinal M2 macrophages, and Il-10 and was correlated with decreased levels of TNF alfa, IL-1β, and RANTES cytokines. Moreover, a *Lactobacillus*-supplemented diet induced an increased functionality of the retinal cones and rods and attenuated the photoreceptor degeneration caused by excessive blue light exposure [68]. 

In the reviewed studies, *Lactobacillus* spp. was found to be decreased in DR patients compared with T2DM patients [19,25], supporting the previous evidence regarding the protective effects of lactic acid bacteria at the retinal level. When compared with HC patients, however, Huang et al. [19] found higher levels of *Lactobacillus* in DR patients, and this may be explained either by the favorable effect of a hyperglycemic environment on the proliferation of these species [25], or by the modulation effect of metformin on gut microbiota [19].

### 4.5. “Leaky Gut” and LPS Endotoxemia

An increase in Bacteroidetes and other pro-inflammatory and mucin-degrading bacteria disrupts the intestinal barrier, allowing bacteria and their metabolites to enter the bloodstream [19,23,25,69]. Bacteroidetes are a vast phylum of gram-negative bacteria that present LPS as a constitutive component of the bacterial outer membrane and are potent triggers for inflammatory response, impaired glucose metabolism, and endothelial dysfunction [19,24,70]. Vagaja et al. [71] reported that systemic LPS exposure in hyperglycemic mice could accelerate the injury of the retinal capillary endothelium and the thinning of the retina. Although the mechanisms of action are not yet fully understood, the evidence to date shows that LPS exhibits apoptotic and neurodegenerative effects upon retinal cells and the EPR and increases the permeability of the blood–retinal barrier, increasing vascular dysfunction and retinal ischemia [70]. LPS induces the conversion of microglia into the M1 pro-inflammatory phenotype via the TLR4 signaling pathway, ROS accumulation, the synthesis of pro-inflammatory cytokines, TNF alfa, and VEGF, neurodegeneration, and gliosis [72]. When exposed to LPS, retinal epithelial cells express high levels of—and secrete a range of—pro-inflammatory cytokines such as IL-6, Il-8, Il-17, IFN-γ, MCP-1, and VEGF, as well as receptors, causing alterations in the endothelial tight junctions and disruption of the blood–retinal barrier [71,72].

### 4.6. Metabolic Changes Related to Gut Dysbiosis in DR

A metabolomic analysis showed that particular changes at the general and species levels observed in patients with DR correlated with either decreases or increases in plasmatic levels of several metabolites which may impact retinal physiology. Nicotinic acid, carnosine, and succinate, which have strong antioxidant properties, were found to be decreased in DR patients compared with HC patients [21,25], making the retina more vulnerable to ROS. Experimental studies have found that dietary supplementation with carnosine can prevent lesions in neurodegenerative disease and diabetic complications [72]. Zhou et al. [21] found a modified arginine–proline metabolic pathway, with a lower level of D-proline in the DR group compared with the T2DM group [21]. In cultured cells, P-proline serves as a nutrient for retinal epithelial cells, promoting cell maturation, increasing reductive carboxylation, and protecting against oxidative damage [73]. In an experimental model, oral supplementation with D-proline improved visual function in patients with age-related macular degeneration [74].

In addition to nutrient deficiency, an increase in toxic and pro-inflammatory metabolites was observed. Traumatic acid resulting from α-linolenic acid metabolism is an activator of caspase 7, inducing cellular apoptosis and causing membrane peroxidation and the accumulation of ROS. Li [25] and Ye [18] have reported higher levels of traumatic acid in DR patients, and this appears to correlate negatively with *Bacillus* species. Li et al. [25] have also observed increased Thromboxane A3 in DR patients when compared with the T2DM group, and this was positively correlated with *Prevotella*. TMAO is a pro-inflammatory metabolite which results from the bacterial oxidation of TMA, a byproduct of the digestion of aliments rich in choline, carnitine, and its derivates. Higher levels of TMAO lead to endothelial dysfunction, oxidative stress, a decrease in anti-inflammatory cytokine Il-10, and increased inflammation [75].

16S rRNA sequencing allows a comprehensive characterization of a specific dysbiosis at the genera and species levels. The study of gut microbiota dysbiosis in diabetic retinopathy has some limitations. The statistical correlations found in the reviewed studies may have been influenced by multiple confounding factors, such as age, diet, sex-related factors, medication, and associated diseases. Once a correlation is proved, there are still questions to be answered, e.g., is dysbiosis a consequence of the disease, or do some specific germs cause or aggravate the retinal damage?

Metformin therapy is one of the most important sources of bias in the reviewed studies due to its correlations with specific changes in gut microbiota which are partially responsible for the therapeutic effect, i.e., the enrichment of SCFA-producing bacteria, such as *Blautia*, *Lactobacillus*, *Bifidobacterium*, *Prevotella*, and *Bacteroides*. In a study on metformin administration in healthy controls, a reduction in the inner diversity of the gut microbiome was observed, together with a relative increase in opportunistic pathogens such as *Escherichia*-*Shigella* spp. and a lower level of Peptostreptococcaceae family bacteria [75]. In an experimental model of high-fat diet-induced T2DM mice, metformin caused a decrease in Clostridia and Proteobacteria, and an increase in Bacilli, Bacteroidetes, and Actinobacteria [76,77].

Animal models have proved that the gut microbiome impacts ocular health, either directly, via intermittent bacteriemia [46], or via their metabolites and signaling molecules. Human studies have evidenced a distinct microbial dysbiosis in DR patients when compared with T2DM patients without DR or with healthy controls. Based on detailed microbiota analyses, researchers have tried to identify specific biomarkers that can be correlated with DR. Bai et al. [22] describe a combination of levels of Blautia, Bacteroides, Megamonas, Romboutsia, and Anaerostipes that could discriminate DR patients from the HC group with an AUC of 0.85 [22]. Liu et al. found that Christensenellaceae, Peptococcaceae, Ruminococcaceae_UCG_011, Eubacterium_rectale_group, and Adlercreutzia effectively discriminate DR and suggest that further studies could confirm whether or not they can be used as novel biomarkers for the early treatment and prevention of DR [27].

A rebalancing of the Firmicutes/Bacteroidetes ratio via diet or probiotics could be an important clinical tool to suppress systemic inflammation, improve insulin sensitivity, and limit retinal damage in diabetes. While Bifidobacterium and Lactobacillus have already been included in clinical trials with favorable results (e.g., promoting intestinal barrier integrity and immunomodulatory effects), other bacteria, such as *Faecalibacterium*, Eubacterium_hallii_group, and *Akkermansia mucinophila*, have been found to be beneficial in experimental studies and may represent possible future directions for clinical research. Verma et al. have developed a probiotic engineering technique, binding human ACE2 with *L. paracasei*, and have successfully reduced the number of acellular capillaries, blocked retinal ganglion cell loss, and decreased retinal inflammatory cytokine expression in two mouse models of diabetic retinopathy [78].

## 5. Conclusions

Diabetic retinopathy is associated with distinctive gut dysbiosis, characterized by an imbalance between pro- and anti-inflammatory species. Furthermore, microbial metabolites may impact retinal nutrition and antioxidant mechanisms. The gut microbiome is extremely diverse and exists in a dynamic co-evolution with the host’s health status, diet, genetic predisposition, hormonal changes, xenobiotics, and aging. The different findings in the published studies included in this review may be seen as pieces of a more complex puzzle, about which there is still much to discover. Further in vivo studies are needed to bring the research data from the benchmark to the bedside and prove the benefits of manipulating microbiota to increase or decrease specific species.

## Figures and Tables

**Figure 1 life-13-00968-f001:**
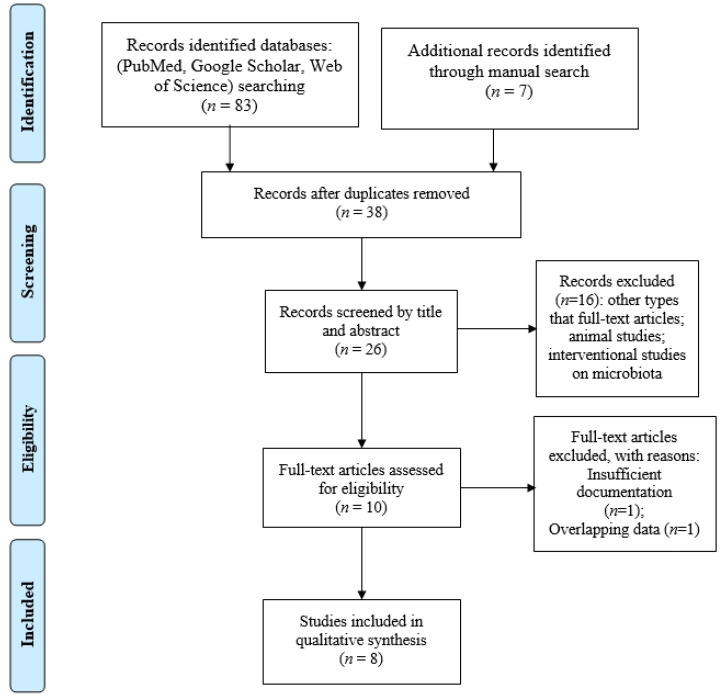
Flowchart of the study selection process in accordance with PRISMA recommendations.

**Table 1 life-13-00968-t001:** Studies on gut microbiota changes in diabetic retinopathy—general data.

Study, Year	Comparison	No. of Patients	Sex Ratio (M/F)	Age (ys ± SD)	BMI (Mean ± SD)	History of DM (ys)	HbA1C (Mean ± SD)	Method of Analysis	Metformin Therapy
Zhou Z [21], 2021	DR	21	14/7	59.57 ± 9.09	22.79 ± 2.43	13 (5–19.5)	6.44 ± 0.92	16S rRNA	21 (100%)
	T2DM	14	8/6	61.93 ± 6.20	22.2 ± 1.65	11.5 (2.7–16.2)	6.55 ± 1.19		14 (100%)
	HC	15	7/8	56.13 ± 8.88	21.23 ± 2.09	-	5.19 ± 1.14		-
Ye P [18], 2021	PDR	45	25/20	59.9 ± 11.3	24.4 ± 2.7	10 (2.5–16.7)	9.6 ± 2.2	16S rRNA	22 (48.8%)
	T2DM (NDR)	90	50/40	60.9 ± 9.9	24.9 ± 3.8	10 (2.0–15.3)	8.8 ± 2.3		48 (53.3%)
Bai J [22], 2022	DR	25	13/12	55.64 ± 6.1	24.11 ± 3.01	10–30	No info	16S rRNA	No info
	HC (spouses)	25	11/14	56.32 ± 6.56	23.73 ± 2.95	-	-		
Huang Y [20], 2021	DR	25 (16 PDR; 9 NPDR)	15/10	60.28 ± 10.5	23.06 ± 2.44	11.69 ± 7.07	No info	16S rRNA	15 (60%)
	T2DM	25	11/14	62.52 ± 7.58	23.83 ± 3.13	10.36 ± 8.08	No info		19 (76%)
	HC	25	9/16	57.80 ± 10.06	24.40 ± 3.51	-	No info		-
Das T [23], 2021	DR (NDPR; PDR)	28 (9;19)	21/7	55.07 (44–69)	No info	No info	No info	16S rRNA	28 (100%)
	T2DM	25	14/11	57.3 (41–71)					25 (100%)
	HC	30	17/13	52.2 (38–81)					-
Moubayed, N. M. [24], 2019	DR	8	0/8	40–60	No info	No info	No info	16S rRNA	No info
	T2DM	9	0/9						
	HC	18	0/18						
Li L [25], 2022	DR	15	8/7	55 (51–63)	26.0 (23.5–28.0)	13 (8–17)	8.7 (7.5–10.4)	Shotgun metagenomic sequencing	No info
	T2DM	15	7/8	57 (51–62)	27.6 (25.5–30.3)	10 (9–14)	7.8 (6.9–9.5)		
Khan R [26], 2021	STDR (PDR; CSME)	37 (21;16)	25/12	57.45 ± 8.08	26.44 ± 5.23	12 (8–20)	7.48 ± 1.44	16s rRNA	4 (10.81%)
	T2DM	21	13/8	57.50 ± 7.60	26.53 ± 5.52	12 (10–20)	7.49 ± 1.48		2 (9.52%)

DR: diabetic retinopathy; NPDR: non-proliferative DR; PDR: proliferative DR; STDR: sight-threatening DR; CSME: clinically significant macular edema; T2DM: type 2 diabetic patients without retinopathy; HC: healthy control; SD: standard deviation; BMI: body mass index; 16s rRNA: DNA extraction and 16S rRNA gene sequencing.

**Table 2 life-13-00968-t002:** Richness and diversity in DR vs. HC.

Study, Year	Chao 1	Sobs	ACE	OTUs	Shannon	Simpson/invSimpson	Heip Evenness Index	Beta Diversity PCoA/PLS-DA
Zhou Z [21], 2021	↑	-	-	↑	Not significant	-	-	Significantly different
Bai J [22], 2022	Not significant	↑	↑ Not significant	-	↑ Not significant	Not significant	Not significant	Significantly different
Huang Y [19], 2021	Not significant	-	Not significant	-	Not significant	↓	↓	Significant diversity
Das T [23], 2021	Not significant	-	-	Not significant	↓	Not significant	-	Significant diversity
Moubayed NM [24], 2019	-	-	-	Not significant	-	-	-	Significant diversity

PCoA: principal coordinate analysis plot; PLS-DA: partial least-squares discriminant analysis; ↑: increased; ↓: decreased.

**Table 3 life-13-00968-t003:** Richness and diversity in DR vs. T2DM.

Study, Year	Chao 1	ACE	OTUs	Shannon	Simpson	Heip Evenness Index	Beta Diversity PCoA
Zhou Z [21], 2021	↑	-	↑	Not significant	-		Significantly different
Ye P [18], 2021	↓	-	↓	↓	↓		Significantly different
Huang Y [19], 2021	Not significant	Not significant	-	Not significant	Not significant	Not significant	Significant diversity
Das T [23], 2021	Not significant	-	Not significant	Not significant	Not significant	-	Significant diversity
Moubayed NM [24], 2019	-	-	Not significant	-	-	-	Significant diversity
Li L [25], 2022	-	-	-	-	-	-	Significant diversity
Khan R [26], 2021	-	-	-	-	-	-	No significant difference in relative abundance; different F/B ratio

↑: significantly increased (*p* < 0.05); ↓: significantly decreased (*p* < 0.05).

**Table 4 life-13-00968-t004:** Genera and species dysbiosis in the reviewed studies.

Phylum	Genera/Species	Main Function	RDM vs. DM	Studies	RDM vs. HC	Studies
Firmicutes	All phylum		↓	Huang Y [19]	↓	Bai, Huang Y [19], Moubayed NM [25]
	*Roseburia* spp.	- Butyrate (SCFA) producer - Anti-inflammatory properties (blocks NDRG2/IL-6/STAT3 signaling pathway) [21,23]	↓	Ye P [18]	↓	Zhou Z [21], Das T [23]
	*Faecalibacterium* spp.	- Butyrate producer - Anti-inflammatory activity NDRG2/IL-6/STAT3 signaling pathway [21] - Metabolic modulator	↓	Ye P [18]	↓	Zhou Z [21], Huang Y [19], Das T [23]
	Ruminococcaceae	- Butyrate producer - Anti-inflammatory role	↓	Zhou Z [21], Ye P [18]	↓	Zhou Z [21]
	*Lachnospira*	- SCFA producers			↓	Das T [23]
	*Dorea*, *Anaerostipes*	- Acetate producers; both positive and negative conditions			↓	Bai J [22]
	*Subdoligranulum*	- Butyrate producer - Mixed effects - Poor metabolism and chronic inflammation [21,29]	↑	Ye P [18]		
	Agathobacter (*Eubacterium rectale*)	- Butyrate producer - Metabolizes lactate - Glutamate metabolism	↑ ↓	Zhou Z [21], Ye P [18]		
	*Veillonellaceae*	- Butyrate producer	↓	Ye P [18]		
	*Streptococcaceae*	- Butyrate producer	↓	Ye P [18]		
	*Clostridium*, Clostridiaceae	- Butyrate producer	↓	Huang Y [19]	↓	Huang Y [19]
	Eubacterium_hallii_group	- Butyrate and propionate production - Anti-inflammatory - Improves insulin sensitivity [19,22]	↓	Huang Y [19]	↓	Bai J [22], Huang Y [19]
	*Blautia*	- TUDCA, TGR5 signaling, preventing DR [19,23]	↓	Huang Y [19]	↑ ↓	Huang Y [19], Das T [23], Bai J [22]
	*Lactobacillus*	- Anti-inflammatory immunomodulatory, antioxidant - Probiotic [19,25,29,30]	↓	Huang Y [19], Li L [25]	↑	Huang Y [19]
	*Lachnoclostridium*	- Acetate and TMA producer - Positive correlation with obesity and T2DM - Pro-inflammatory [22]			↑	Bai J [22]
	*Romboutsia*	- SCFAs producer, anti-inflammatory			↑	Bai J [22]
	Megamonas	- Acetate producer, anti-inflammatory, lipogenesis			↑	Bai J [22], Das T [23]
	Ruminococcus_torques_group	- Mucin degrading [31,32]			↑	Bai J [22]
	Peptostreptococcaceae	- SCFA producer, anti-inflammatory			↓	Huang Y [19]
	Oscillospiraceae	- SCFAs - Both positive and negative effects [33,34]	↑	Huang Y [19]		
	Christensenellaceae	- SCFA producer - Co-occurrence with methanogens - Anti-inflammatory [35,36,37]	↑	Huang Y [19]		
	Acidaminococcaceae	- Glutamate degrading - Pro-inflammatory - Altered amino acid metabolization and gut permeability [38,39,40]	↑	Huang Y [19], Das T [23]		
Actinobacter	All phylum		↓	Li L [26], Das T [23]	↓	Das T [23]
	*Coriobacteriales*	- Glucose homeostasis via liver energy metabolism - Glutamate metabolism [18]	↓	Ye P [18]		
	*Collinsella*	- Butyrate producer			↓	Bai J [22]
	*Bifidobacterium*	- Improves glucose tolerance, metformin related [23,25]	↓	Das T [23], Li L [25]	↓ ↑	Das T [23], Huang Y [19]
	*Atopobiaceae*	- lactate and butyrate producer - bile acid degrader [41]	↑	Huang Y [19]	↑	Huang Y [19]
Bacteroidetes	All phylum	- LPS endotoxemia, systemic inflammation, vascular dysfunction; endothelial cell damage [22]	↑	Huang Y [19], Li L [25]	↑	Bai J [22], Huang Y [19], Moubayed NM [24]
	*Prevotella*	- Systemic inflammation, TLR2 activation, IL-17A; IL-17R-Act1-Fas-activated death domain (FADD)-axis induced endothelial damage [21,42]	↑	Zhou Z [21],	↑	Moubayed NM [24]
	Parabacteroides	- SCFAs producer - Anti-inflammatory (TUDCA producer) [43] - Pro-inflammatory (via succinate and HIF-1alfa) [23]			↑	Das T [23]
	Muribaculacea	- LPS endotoxemia - Systemic inflammation, TLR 4 activation, ↑TNF-α, and IL-6 - ↑TMAO [44]	↑	Huang Y [19]	↑	Huang Y [19]
	Bacteroides	- Inflammation, LPS endotoxemia			↑	Bai J [22], Das T [23]
	Alistipes	- Inflammation			↑	Li L [25], Bai J [22], Das T [23]
Proteobacteria	*Burkholderiaceae*	- Systemic inflammation, chemokine IP-10	↑	Ye P [18]		
	*Morganella*	- Glutamate metabolism	↓	Ye P [18]		
	*Pasteurellaceae*	- LPS bacteria; some spp are pathogens	↓	Huang Y [19]	↓	Huang Y [19]
	*Escherischia*	- Pathogen	↑	Das T [23]	↑	Das T [23]
	*Enterobacter*	- Pathogen	↑	Das T [23]		
Verrucomicrobia	All phylum				↑	Bai J [22]
	*Akkermansia muciniphila*, Akkermansiaceae	- Mucin-degrading bacteria, intestinal barrier disruption; [21,27] - Increases insulin sensitivity and prevents fatty liver [21,45,46] - Anti-inflammatory [23]	↑	Das T [23]	↑	Zhou Z [21], Das T [23]
*Desulfobacterota*	All phylum	- Reduces sulfur compounds via the DsrAB-dissimilatory sulfite reduction pathway [19,22] - Butyrate degradation via the butyrate beta-oxidation pathway [19,22]	↑	Huang Y [19]	↑	Bai J [22]
Synergistota	All phylum	- Imbalance in catabolic reactions, energy metabolism [22]			↑	Bai J [22]
F/B ratio			↓ Not significant	Khan R [26], Huang Y [19], Ye P [18]	↓	Bai J [22], Huang Y [19], Moubayed NM [24]

↑: significantly increased (*p* < 0.05); ↓: significantly decreased (*p* < 0.05)

## Data Availability

No new data was created.

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
