# Peer review of "Gut Microbiota Dysbiosis in Diabetic Retinopathy—Current Knowledge and Future Therapeutic Targets"

_life, 2023, doi:10.3390/life13040968_

Round 1
Reviewer 1 Report
The review of Serban et al investigated in details, starting from the recent literature, the interaction between gut microbiota and diabetic retinopathy in individuals affected by type 2 diabetes. The results of this meta-analysis confirmed the existence of significant microbial differences between diabetic patients with and without retinopathy. The Authors conclude that gut microbiota dysbiosis could represent a novel therapeutic target for the prevention/treatment of diabetic retinopathy.
Comments to the manuscript are:
This is a well organized and well written review based on updated literature. The conclusions are supported by the results obtained and, altogether, the results may actually add to the present knowledge in the field.
Author Response
Dear Reviewer,
Thank you very much for your kind comments and appreciation of our work!
Reviewer 2 Report
In their meta-analysis, D. Serban and colleagues report about “Gut microbiota dysbiosis in diabetic retinopathy – current knowledge and future therapeutical targets”. It is readily evident that the expert authors wish to provide interesting insights into this extremely complex field. Limiting for this undertaking is the paucity of nine partially contradictory manuscripts. The resulting set of confusing findings does not help the reader of their manuscript to depict the key messages, while obviously also the authors themselves are somewhat lost. This becomes obvious with a long list of microbiologic details and their impact on ocular health and disease, which cannot readily be translated into clinical conclusions. As a non-microbiologist I would strongly suggest to shorten the discussion 8i.e. chapter 4.1) by two thirds and limit it to information that is robust, independently confirmed and relevant in a clinical context with focus on translational measures or to choose a journal with stronger focus on gut microbial pathology. Further comments:
Statistics have not been described in methodology.
How was confirmed that no overlapping data were included (.i.e. Jayasudha 2020 and Das 2021)?
Line 158: 6/9 studies in the text is not in line with 5/9 in table 1…
Line236: What is “enhanced inflammatory status”? increase in CRP? Specific microbiota changes?
Line 243-5: … or be the consequence of the underlying metabolic pathology.
Line 257: reference missing
Taken together, interesting information, difficult drain from the current manuscript.
Author Response
Dear Reviewer,
Thank you very much for your professional help and time spent reviewing our paper. We agree to all your comments and recommendations and we have carefully revised our manuscript accordingly.
We have shortened the 4.1 chapter, according to your recommendations, and focused on the most relevant data.
Further comments:
Statistics have not been described in methodology.
We added a paragraph regarding the statistic analysis of the studies included in the present systematic review.
How was confirmed that no overlapping data were included (.i.e. Jayasudha 2020 and Das 2021)?
Thank you for the observation. Initially, we included both the studies of Jayasudha and Das, due to the fact that Jayasudha et al reported the changes in the patients’ micobiome (and it was the only study analyzing fungi that we have found in the databases), while Das et al analyzed changes in microbiome. However, we agree that there might be overlapping data, based on the similar general characteristics of the healthy controls (age, sex distribution) and the methodology of research of those 2 authors.
Considering also your recommendation to focus on the most robust and relevant information on the subject, we have decided to exclude the study of Jayasudha from the analysis and we have corrected the data and the references accordingly.
Line 158: 6/9 studies in the text is not in line with 5/9 in table 1…
Thank you for the observation. We have corrected
Line236: What is “enhanced inflammatory status”? increase in CRP? Specific microbiota changes?
We have rephrased it. What we meant is to point out the imbalance between pro and anti-inflammatory bacteria in the gut microbiome.
Line 243-5: … or be the consequence of the underlying metabolic pathology.
We have added this idea. Thank you for the suggestion!
Line 257: reference missing
We have rephrased the paragraph and added the appropriate references.
We have also carefully checked our manuscript for English spelling and grammar, as recommended.
We hope in this revised version, you will find our paper suitable for publication.
Reviewer 3 Report
The article entitled: Gut microbiota dysbiosis in diabetic retinopathy – current knowledge and future therapeutical targets is a systematic review very well written. Data presented by the authors are clear and concisely. The theme is important in clinical practice especially because is about therapeutically targets. Discussions are centered on multiple SCFAs producing bacteria that can be used as a potential therapeutic target due to their beneficial effects on the host.
Author Response

(The authors gave the same response as above.)

Round 2
Reviewer 2 Report
I think the authors improved their manuscript satisfyingly.